# Human encroachment, climate change and the loss of our archaeological organic cultural heritage: Accelerated bone deterioration at Ageröd, a revisited Scandinavian Mesolithic key-site in despair

Adam Boethius[1]*, Mathilda Kjällquist[2], Ola Magnell[2], Jan Apel[3]

1 Department of Archaeology and Ancient History, Lund University, Lund, Sweden, 2 The Archaeologists, National Historical Museums, Lund, Sweden, 3 Department of Archaeology and Classical Studies, Stockholm University, Stockholm, Sweden

* adam.boethius@ark.lu.se

**Data Availability Statement:** All relevant data are within the paper and its Supporting Information files.

## Abstract

Ancient organic remains are essential for the reconstruction of past human lifeways and environments but are only preserved under particular conditions. Recent findings indicate that such conditions are becoming rarer and that archaeological sites with previously good preservation, are deteriorating. To investigate this, we returned to the well-known Swedish Mesolithic site Ageröd I. Here we present the result of the re-excavation and the osteological analyses of the bone remains from the 1940s, 1970s and 2019 excavation campaigns of the site, to document and quantify changes in bone preservation and relate them to variations in soil conditions and on-site topography. The results indicate that the bone material has suffered from accelerated deterioration during the last 75 years. This has led to heavily degraded remains in some areas and complete destruction in others. We conclude that while Ageröd can still be considered an important site, it has lost much of the properties that made it unique. If no actions are taken to secure its future preservation, the site will soon lose the organic remains that before modern encroachment and climate change had been preserved for 9000 years. Finally, because Ageröd has not been subjected to more or heavier encroachment than most other archaeological sites, our results also raise questions of the state of organic preservation in other areas and call for a broad examination of our most vulnerable hidden archaeological remains.

## Introduction

Researchers working with previously excavated archaeological sites have noticed that prehistoric organic remains are more rarely recovered today and that bones from earlier excavations are generally better preserved. This is tacit yet wide-spread knowledge, but the phenomenon has rarely been properly analysed, reported or measured [1, 2]. As a first step to remedy this,

**Funding:** We are grateful for the financial support from: JA, Crafoord foundation, nr. 20180631 https://www.crafoord.se/ AB, Lennart J. Hägglunds foundation for archaeological research and education http://hagglundsstiftelse.se/ AB, the Swedish National Heritage Board, RAÄ-2018-3237, https://www.raa.se/ The funders had no role in study design, data collection and analysis, decision to publish, or preparation of the manuscript.

**Competing interests:** The authors have declared that no competing interests exist.

in May 2019 we conducted a small scale excavation campaign at the Middle Mesolithic site Ageröd I:HC, located in central Scania southern Sweden. Here, the faunal remains are analysed and compared with the remains recovered on previous excavation campaigns at the site, to measure the rate of deterioration over the last 75 years.

There are few published reports concerning the ongoing soil acidification and its effect on non-excavated organic components on archaeological sites. However, in Sweden, around three decades ago, concerns were raised regarding the effect of modern pollution on other types of cultural heritage remains. As a consequence of these concerns, the Swedish National Heritage Board initiated a project in 1988 to investigate how air pollution affected ancient monuments and buildings [3, 4]. Part of this investment was also diverted into investigating archaeological artefacts from earthbound contexts, focusing on metal objects. A correlation between modern soil acidification and accelerated metal corrosion during the last 50–100 years was demonstrated [5]. Following this, investigations on archaeological materials in different soils were expanded and came to include bone deterioration and evaluations of different soils in selected European countries [6]. The results indicated that the archaeological record was threatened and that in some areas organic material and non-noble metals faced destruction [5].

While indications of ongoing destruction and threatened archaeological material were evident, limited follow-up investigations were conducted. Also, most of the efforts from these studies focused on showing the present state of the archaeological objects and on differences in site-specific variations in bone degradation (e.g. [7]) or information on how to study and detect deterioration in different types of source materials [8]. Furthermore, site-specific deterioration in progress (i.e. comparison with museum collections) has been identified as an important aspect in furthering the information regarding the preservation of our ancient heritage [9]. In 2011, Nicky Milner *et al* could demonstrate alarming organic deterioration on a site-specific level of the Mesolithic site Star Carr, by comparing excavated materials from different time periods [10, 11]. The results from the Star Carr excavation, in combination with a growing body of evidence of accelerated organic degradation [5–7, 9–15], suggest that more data on the status of ancient organic remains is needed. Furthermore, a general decline of wetlands (through drainage) in Europe is resulting in accelerated peat and gyttja degradation; which, because the anoxic peat layers in a natural unaltered peat bog have for millennia protected the archaeological materials (see e.g. [16, 17]), calls for new investigations about the wellbeing of the organic parts of our common legacy and new means of quantifying contingent ongoing degradation of organic material at sites previously known for their good preservation.

## The Ageröd site

Ageröd I is an Early Holocene site, Late Maglemose to Early Kongemose culture using south Scandinavian chronology [18], dated to roughly 8700–8200 cal BP. Ageröd I is located by a peat bog, which was once a shallow lake. The Ageröd peat bog is located the north of Lake Ringsjön in mid-Scania, southern Sweden (Fig 1). This area is known for its rich Mesolithic sites with organic remains [19–31].

Ageröd I was discovered in the 1930s [32]. However, due to the Second World War, it took until 1946–1949 until Carl-Axel Althin at Lund University Historical Museum excavated the site (Fig 2). During the following ambitious, detailed and extensive excavations the site was divided into five major sections: A, B, C, HC, and D (see Fig 3 in [33]). Ageröd I consists of three settlements with section A, C and HC as representing one settlement and Ageröd I:B and Ageröd I:D as two others [21]. Althin recovered numerous finds and it soon became apparent

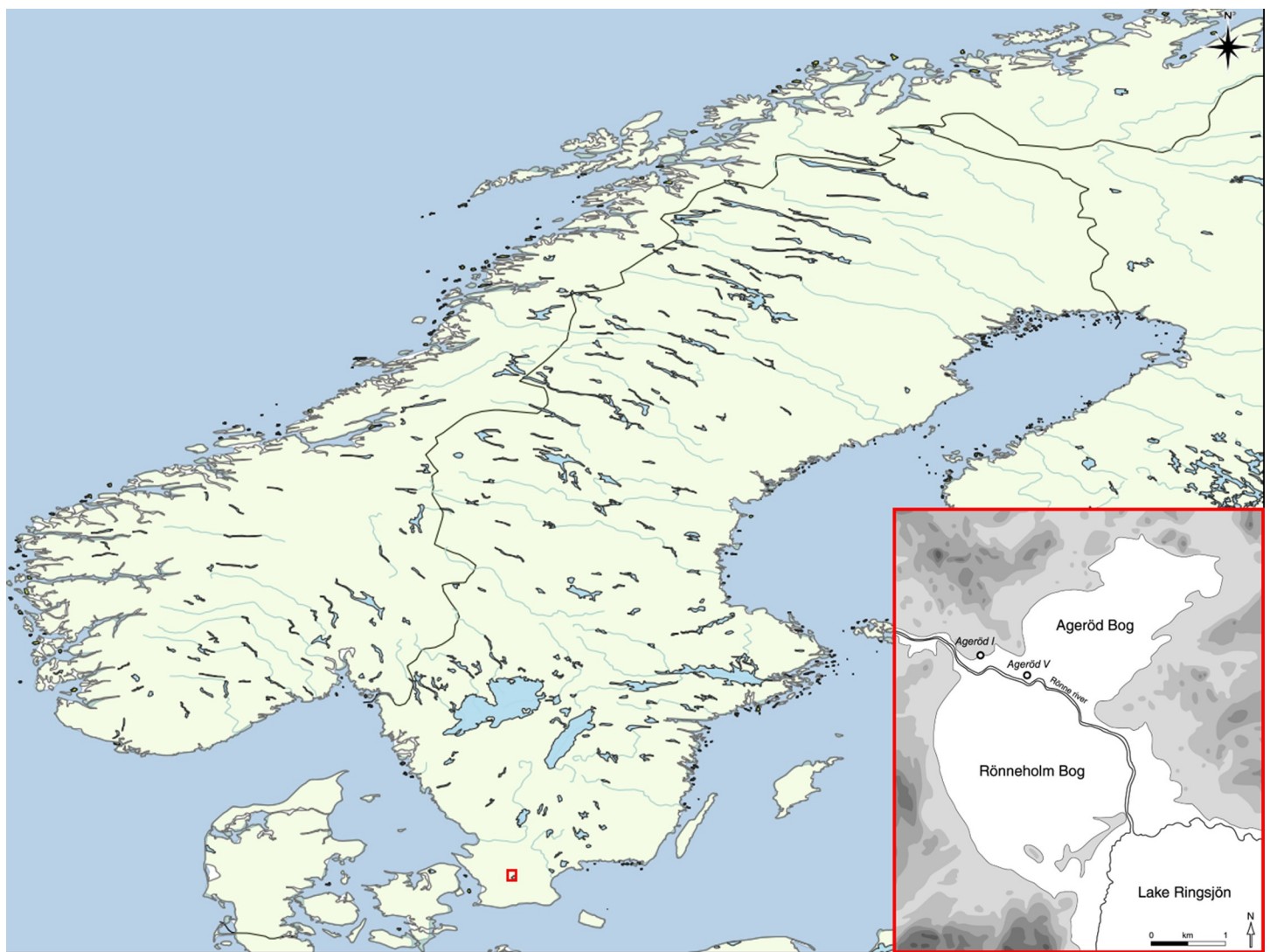

**Fig 1. Map of Scandinavia zoomed in on the area of the Ageröd I site located on the ancient shore of the former shallow lake.** World map generated with QGIS 3.10 using the Natural Earth data set. Lower right drawing of the Ageröd area by Arne Sjöström. Image created for this publication.

that the first well-preserved Middle Mesolithic site in Sweden displaying phenomenal preservation and large quantities of both bone materials and flint had been recovered [34].

New excavations were carried out by Lars Larsson between 1972 and 1974. These investigations allowed Larsson to assess settlements patterns, chronology, seasonality, population size and resource exploitation [21] and present an overview of the bone and antler artefacts from the site [35]. Apart from the previous excavations on the site (Fig 3), the Ageröd bog has also gone through extensive quaternary geological analyses to establish a thorough stratigraphy of the bog, create pollen analytical dating sequences and to map the formation of the bog and its ancient shorelines [33, 36, 37].

Most of the organic finds from Ageröd I consist of bone and antler refuse. However, while osteological analyses were conducted on the bones recovered from the Ageröd I: B & D sections in the 1970s-excavations and published as a summary in the appendix to Larsson´s dissertation [38], the majority of the bones (from the I:HC section) have remained unpublished.

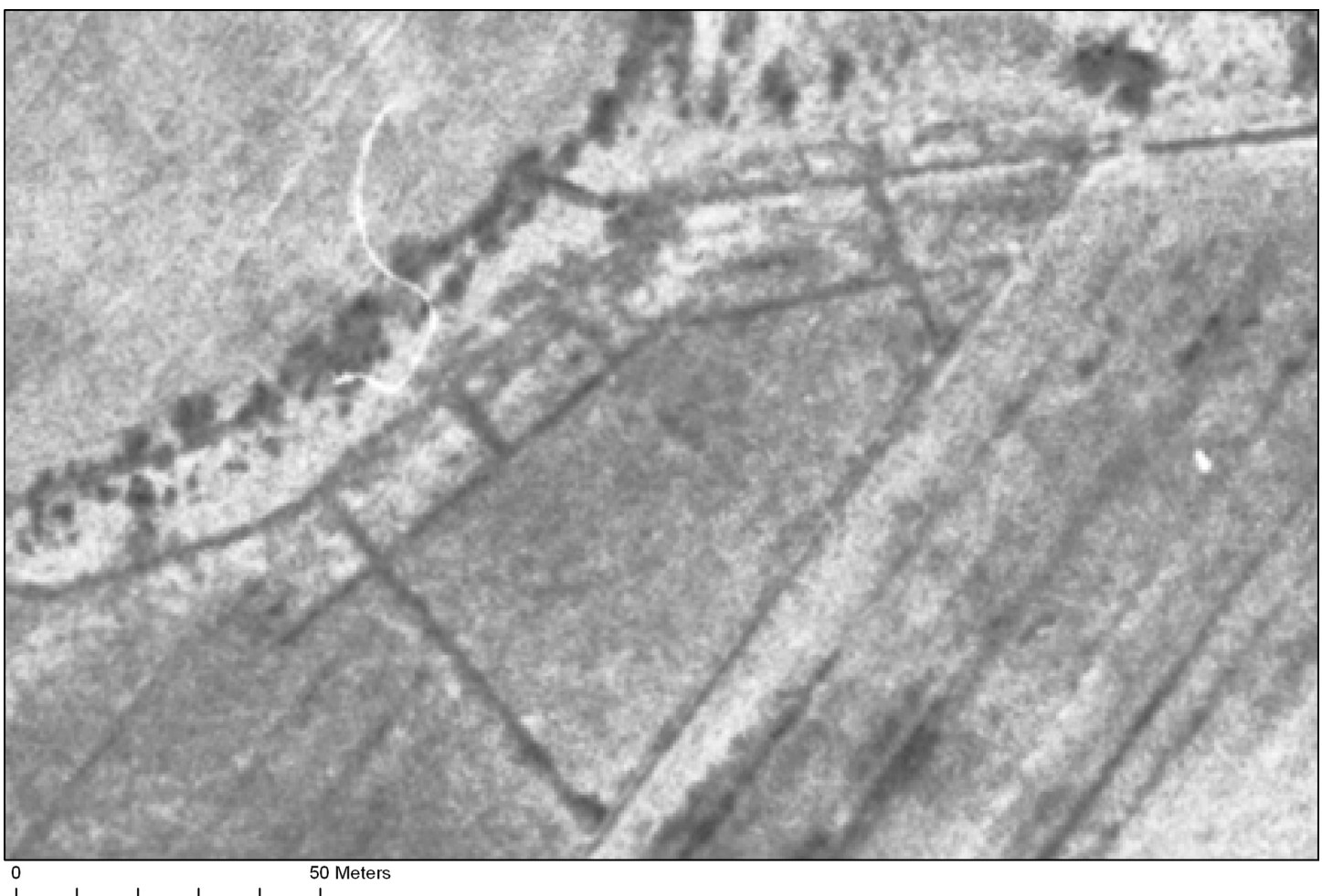

**Fig 2. Aerial photography from 1957 (freely available from Lantmäteriet through CC by 0 license) showing the original trenches from the 1940s.**

The bones from previous excavations of the site have now been analysed (I:HC) or re-analysed (I:B, D) and are briefly published here (Table 1) along with the zooarchaeological analysis from the 2019 excavation campaign. The material from previous excavations of Ageröd I (B, D & HC) amounts to a total of 4180 bone fragments determined to species or family level. From the Ageröd I:HC section 299 bone or antler artefacts have also been identified [35], which combined with the 18 bone and antler artefacts from the I:B & D section [21] make the bone and antler assemblage from Ageröd I one of the largest and best-preserved Middle Mesolithic assemblages in northern Europe. When also considering other organic remains, recovered on the early excavations of the site, such as worked wood, resin, pollen and hazelnuts etc., the vast potential of the organic material culture from the site is highlighted.

Today, the area around Ageröd has been turned into farmland. To make the bog arable, farmers have, since the beginning of the 20th century, been digging ditches around and straight through the bog for drainage (Fig 3A). It is safe to assume that the Ageröd bog started drying up with the digging of the initial drainage ditches. Indeed, compaction and degradation of the peat layers were noted already in the 1970s. Between 1948 and 1973, observations of a diminished peat thickness of more than 10 cm were made when comparing the documentation of the peat layers from the different excavations ([21], 152).

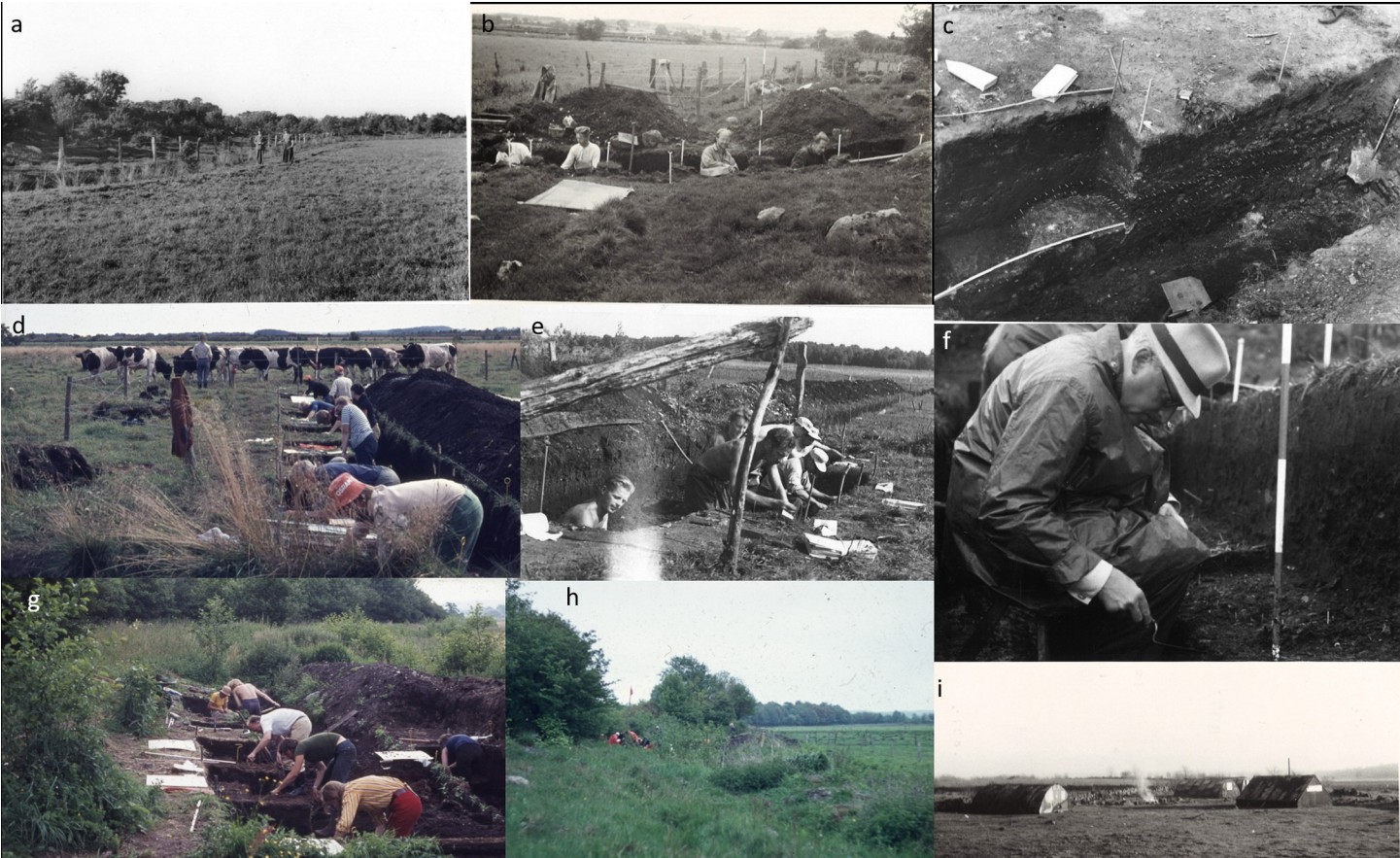

**Fig 3.** Photographs from the 1940s and the 1970s excavation campaigns a) Surveying Ageröd I field prior to 1940s excavation, note fenced-in drainage ditch already in place. b) 1940s excavation in Ageröd I:HC. c) 1940s trench section with detailed separation of different layers. d) 1970s excavation in Ageröd I. e) 1940s excavations under a sunshade in Ageröd I. f) Zooarchaeologist Herved Berlin excavating in Ageröd I:HC in the 1940s. g). 1970s excavation in Ageröd I, h) 1970s area topography of Ageröd I with the ongoing excavation in the background. i) 1940s excavation campaign camp facility setup. Photography: 1940s photos scanned by the authors from original photos deposited at Lund University Historical Museum and made available through CC by 4.0 license; 1970s photos by Lars Larsson are freely available. Picture collection created for this publication by the authors (AB).

The location of the Ageröd I site is fairly secluded and no roads, railroads or even modern houses have been constructed near the site. Consequently, the major disturbances to the sites come from background pollution (acidic precipitation, exhaust etc.), climate change (larger fluctuation in groundwater levels due to warmer summers), the ditches draining the bog, ploughs turning the upper peat layers and previous excavations at the site.

## The excavation

Ageröd was revisited between the 16$^{th}$ and 24$^{th}$ of May 2019 to investigate the *in situ* status of the organic material at the site. Because further excavations of the site may contribute to the destruction of organic material, our intrusions were kept to a minimum and five test pits were excavated by hand in the close vicinity of the previous excavations at Ageröd I:HC (Fig 4). The excavation was carried out by Lund University and The Archaeologists, National Historical Museums in Sweden. During the excavation campaign, 22 different researchers and archaeologists from 13 research institutes and universities visited and participated.

The test pits were located close to areas where most organic remains had been recovered during previous excavations. The absolute location of the original trenches was determined by

**Table 1. Animal species distribution (NISP) from all excavation campaigns at Ageröd I.**

| Family | Species | Ageröd I: HC 1940s | Ageröd I:HC 1970s | Ageröd I:HC 2019 | Ageröd I:HC Combined | Ageröd I:B 1940+1970s | Ageröd I:D 1940+1970s excavations | Ageröd I Total |
|---|---|---|---|---|---|---|---|---|
| **Mammals (Mammalia)** | | | | | | | | |
| Cervidae | Red deer (*Cervus elaphus*) | 1426 | 62 | 22 | 1510 | 72 | 123 | 1705 |
| | Roe deer (*Capreolus capreolus*) | 326 | 20 | 7 | 353 | 18 | 12 | 383 |
| | Elk (Moose in North America) (*Alces alces*) | 219 | 20 | 5 | 244 | 10 | 11 | 265 |
| | Cervidae indet. | 79 | 1 | 3 | 83 | 42 | 3 | 128 |
| Bovidae | Aurochs (*Bos primigenius*) | 265 | 28 | 2 | 295 | 5 | 17 | 317 |
| | *Aurochs (*Bos primigenius*)/ European bison (*Bison bonasus*) | 51 | | | 51 | 1 | | 52 |
| | Aurochs (*Bos primigenius*)/ Cattle (*Bos taurus*) | 9 | | | 9 | | | 9 |
| | Cattle (*Bos taurus*) | 1 | | | 1 | | | 1 |
| Suidae | Wild boar (*Sus scrofa*) | 848 | 64 | 12 | 924 | 54 | 34 | 1012 |
| Equidae | Horse (*Equus caballus*) | 1 | | | 1 | 5 | | 6 |
| | Large ungulate | 17 | | 6 | 23 | | | 23 |
| Phocidae | Grey seal (*Halichoerus grypus*) | 2 | | | 2 | | | 2 |
| | Harp seal (*Pagophilus groenlandica*) | 1 | | | 1 | | | 1 |
| | Phocidae indet. | 5 | | | 5 | | | 5 |
| Ursidae | Brown bear (*Ursus arctos*) | 62 | 3 | 2 | 67 | 3 | 2 | 72 |
| Canidae | Wolf (*Canis lupus*) | 4 | | | 4 | | | 4 |
| | Red fox (*Vulpes vulpes*) | | | | 0 | 2 | | 2 |
| | Dog (*Canis familiaris*) | 40 | 1 | | 41 | 9 | 18 | 68 |
| | Canidae indet. | 5 | | | 5 | 1 | 1 | 7 |
| Mustelidae | Badger (*Meles meles*) | 11 | 1 | | 12 | 1 | 2 | 15 |
| | Otter (*Lutra lutra*) | 4 | | | 4 | | | 4 |
| | Pine marten (*Martes martes*) | 11 | | | 11 | | 2 | 13 |
| | European polecat (*Mustela putorius*) | 1 | | | 1 | | | 1 |
| | Badger (*Meles meles*) / Otter (Lutra lutra) | 1 | | | 1 | | | 1 |
| Felidae | Wild cat (*Felis silvestris*) | 3 | | | 3 | | 1 | 4 |
| | European lynx (*Lynx lynx*) | 1 | | | 1 | | | 1 |
| | Carnivora indet. | 3 | | 1 | 4 | 1 | | 5 |
| Erinaceidae | European hedgehog (*Erinaceus europaeus*) | 2 | | | 2 | | | 2 |
| Castoridae | Beaver (*Castor fiber*) | 67 | 1 | 1 | 69 | 8 | 2 | 79 |
| Sciuridae | Red squirrel (*Sciurus vulgaris*) | 3 | | | 3 | | | 3 |
| Hominidae | Human (*Homo sapiens*) | 7 | | | 7 | | | 7 |
| NISP | **Sum of identified mammals** | 3475 | 201 | 61 | 3737 | 232 | 228 | 4197 |
| **Birds (Aves)** | | | | | | | | |
| Anatidae | Northern shoveler (*Anas clypeata*) | 2 | | | 2 | | | 2 |
| | Eurasian wigeon (*Anas penelope*) | 1 | | | 1 | | | 1 |
| | Northern pintail (Anas acuta) | 2 | | | 2 | | | 2 |
| | Red-breasted merganser (*Mergus serrator*) | 1 | | | 1 | | | 1 |
| | Bean goose (*Anser fabalis*) | 1 | | | 1 | | | 1 |
| | Brent goose/Barnacle goose (Branta bernicla/Branta leucopsis) | 1 | | | 1 | | | 1 |

(*Continued*)

**Table 1.** (Continued)

| Family | Species | Ageröd I: HC 1940s | Ageröd I:HC 1970s | Ageröd I:HC 2019 | Ageröd I:HC Combined | Ageröd I:B 1940+1970s | Ageröd I:D 1940+1970s excavations | Ageröd I | Total |
|---|---|---|---|---|---|---|---|---|---|
| | Common goldeneye/tufted duck (Bucephala clangula/Aythya fuligula) | 1 | | | 1 | | | 1 | |
| | Barnacle -/Greater white-fronted goose (Branta leucopsis/Anser albifrons) | 1 | | | 1 | | | 1 | |
| | Anserini indet. | 3 | | | 3 | | | 3 | |
| | Anatidae indet. | 3 | | | 3 | | | 3 | |
| | Cygnus sp. | 3 | | | 3 | | | 3 | |
| Podicipedidae | Great crested grebe (*Podiceps cristatus*) | 3 | | | 3 | | | 3 | |
| | Grebe indet. (Podiceps sp.) | 1 | | | 1 | | | 1 | |
| Rallidae | Common moorhen (*Gallinula chloropus*) | 1 | | | 1 | | | 1 | |
| Phasianidae | Western capercaillie (*Tetrao urogallus*) | 5 | | | 5 | | | 5 | |
| | Hen harrier (*Circus cyaneus*) | 1 | | | 1 | | | 1 | |
| | White-tailed eagle (*Haliaeetus albicilla*) | | | | 0 | | 1 | 1 | |
| | Aves indet. | 2 | | | 2 | | | 2 | |
| NISP | **Sum of bird specimens** | 32 | | 0 | 32 | 0 | 1 | 33 | |
| **Fish (Pisces)** | | | | | | | | | |
| Cyprinidae | Tench *(Tinca tinca)* | | | | 0 | | 1 | 1 | |
| Esocidae | Pike (Esox lucius) | | 1 | | 1 | 1 | 1 | 3 | |
| NISP | **Sum of identified fish** | 0 | 1 | 0 | 1 | 1 | 2 | 4 | |
| **Reptiles (Reptilia)** | | | | | | | | | |
| Emydidae | Emys orbicularis | 6 | | | 6 | | | 6 | |

using rectified aerial photographs from 1957, where the trenches could be seen (Fig 2). This data was converted into the modern national grid system. It was tested for accuracy in the field by locating specific fix points hammered and drilled into large stones at the site on previous excavations, and by discussing the fix points from the 1970s with professor Larsson visiting the site.

To facilitate interpretations, the Ageröd I:HC area was divided into four zones. This included two zones designed to cover the area to the north of the added soil bank, zone 1 with original X-values between -9 to -14 and zone 2 encompassing the area of the added soil heap, with original x-values between -6 to -8. Two zones were also created to encompass the area on the lakeside of the ditch, zone 3 encompasses the first "shallow" parts of the former lake, with original x-values between -5 to -0, zone 4 encompasses the "deep" parts of the former lake including only positive x-values (+0 to +50).

Osteological analyses on the original osseous material, related to the rectified GIS-data, enabled quantifications of the number of bones found in each square meter on previous excavations. This ensured that the new test pits were placed in the areas where the best-preserved and largest quantities of bones had been recovered on previous excavations.

During the excavation, five 1x1 meter trenches were excavated. In four of the five trenches, the archaeological deposits were intact. In one case, the new trench (259) was located partly within a previously unpublished and unknown trench. Consequently, the results from this trench are biased and excluded in the following discussions. The trenches were excavated

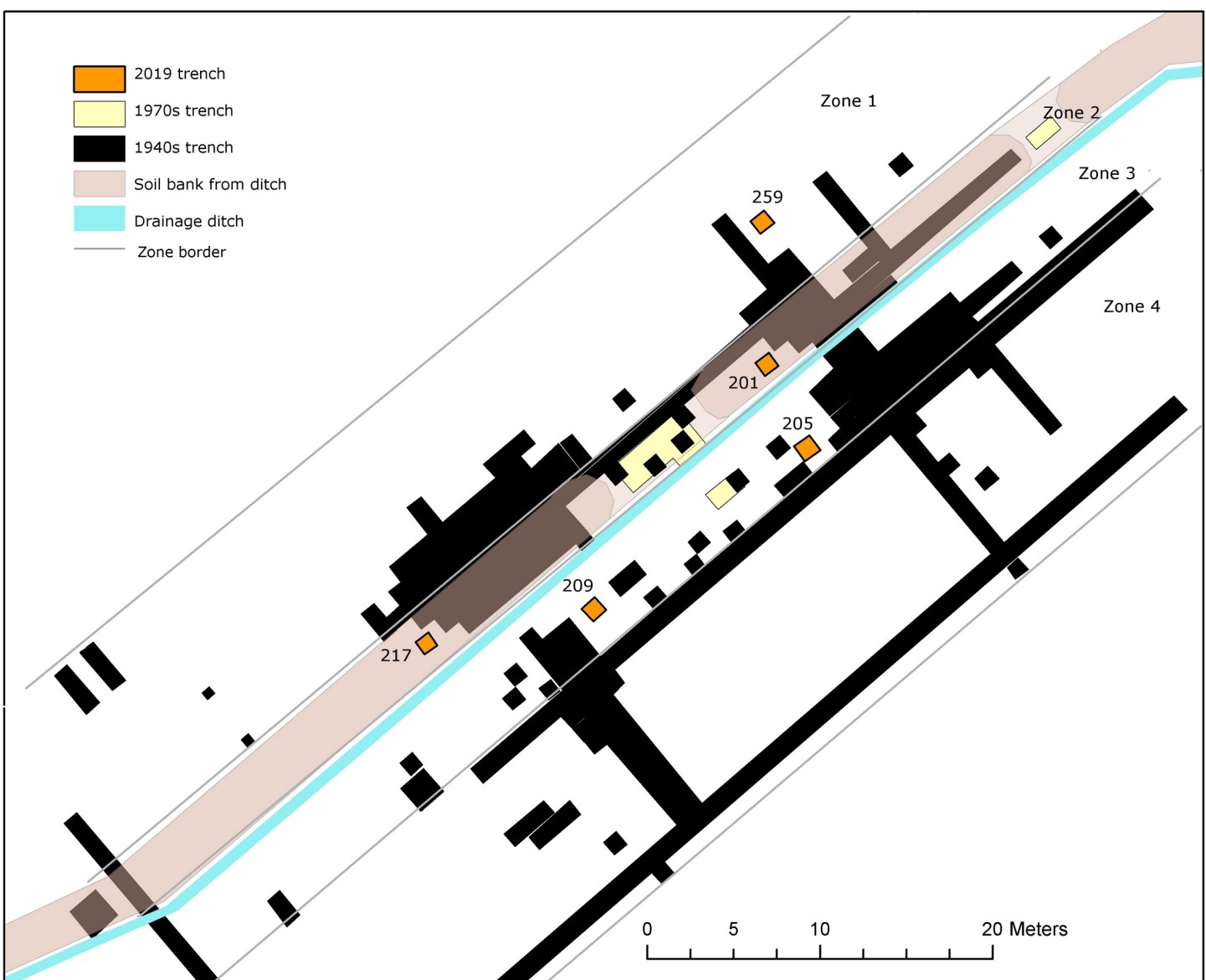

**Fig 4. The square meter trenches from 2019 in relation to the former excavation trenches, the ditch draining the bog and the soil bank of excavated ditch material from its establishment and maintenance.** The zone division was arbitrarily selected to study differences in bone preservation between the driest (zone 1) and the wettest (zone 4) conditions.

stratigraphically, and all archaeological layers (Fig 5) were sampled for soil analyses and macrofossils. The results from the soil and macrofossil analyses will be published separately, together with histological analyses [39]. To ensure maximum possible recovery and to avoid missing small fragments, two-step water sieving (5 mm mesh placed on top of a 2,5 mm mesh) was used. All soil from the major archaeological layers in trenches 201 and 217 was water sieved while the soil from the more peripheral layers was sampled and between 15–35% of that soil was water sieved (S1 Table in S1 File).

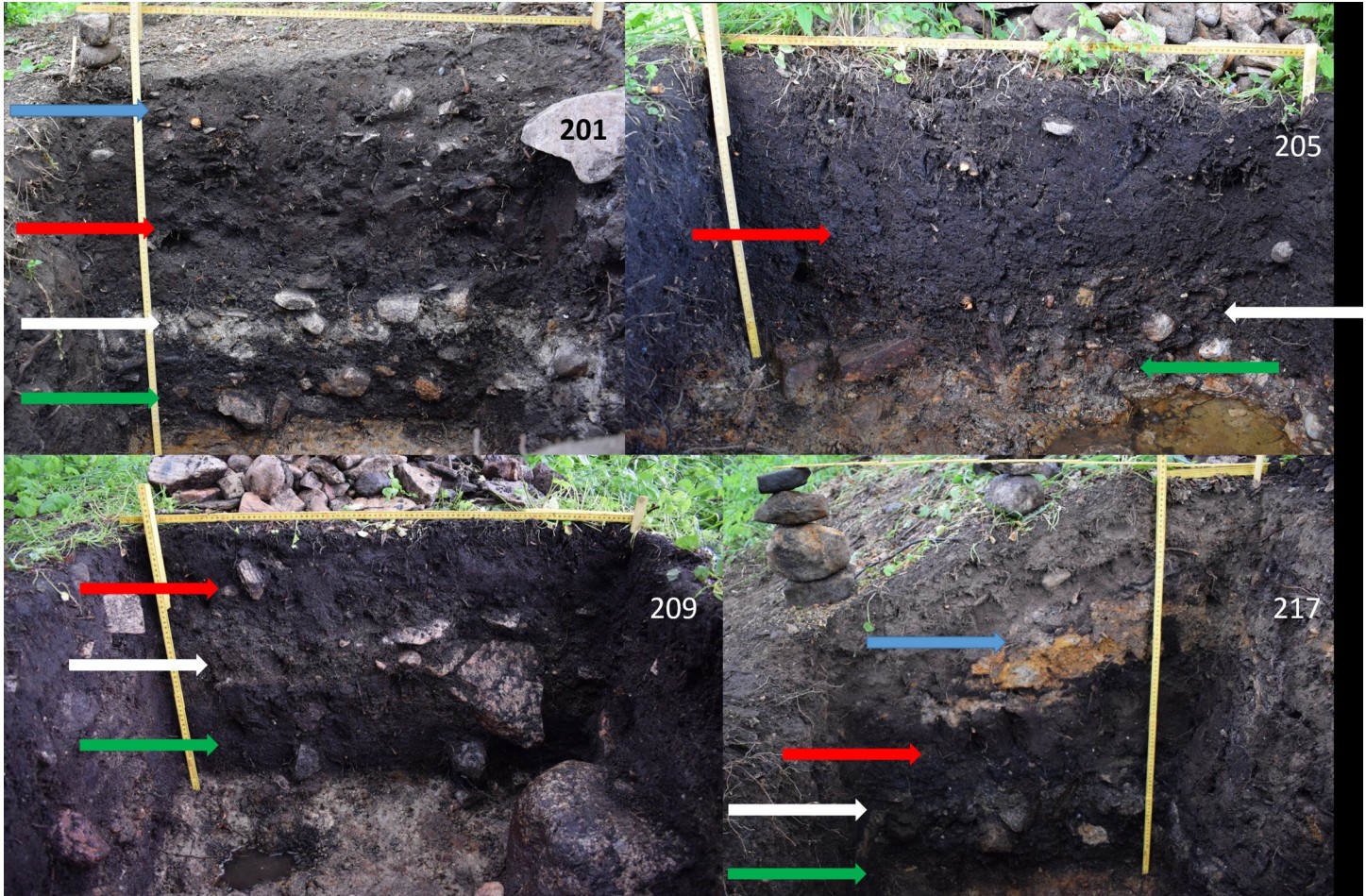

**Fig 5. Sections from the four undisturbed trenches excavated in 2019.** Blue arrow shows added soil from the drainage ditch, red arrow shows the upper peat layer, white arrow shows the white archaeological cultural layer and green arrow shows the lower peat layer. The different layers are most evident in trench 201, in trench 205 the lower peat layer is almost completely gone and difficult to detect as it has deteriorated and become merged with the white cultural layer. Trench 217 has the most amounts of added soil from the drainage ditch on top of the originally deposited layers and the ditch has been dug down into the moraine, as shown with the orange morainic soil in the added soil layer. Image created for this publication by the authors (MK and AB).

## Methods

### Permissions

The 2019 excavation of Ageröd was conducted with permission from the County Administrative Board, Scania, Sweden (reference number 431-3998-2019), in accordance with Swedish legislation. The permit allowed excavation with minor intrusions to meet the specified aims of investigating the preservation status at the site and it allowed follow up analyses on the recovered remains (both destructive and non-destructive) to generate quantitative and qualitative data on the organic deterioration. The archaeological remains recovered on the excavation are temporarily stored at the The Archaeologists, National Historical Museums, Lund, Sweden, but will be transferred to Lund University Historical Museum, Lund, Sweden (LUHM), where they will be permanently deposited along with the remains from previous excavation campaigns at Ageröd. The zooarchaeological analyses conducted on the bone remains from the 1940s and 1970s excavation campaigns were done with permission from LUHM. The remains are stored at LUHM under the deposition numbers LUHM 28977 and LUHM 80910, and are

available at the museum upon request. An overview of all analysed specimens is presented in the Material section below. The permission obtained from LUHM also included the sampling of six bones to conduct histological analyses (ca 3 g of bone per sample) and an additional 12 bone samples (0,5–1 g each) to investigate collagen deterioration (decision 29-08-2019). The results from these analyses will be published separately [39].

## Zooarchaeological analysis

Renewed analyses of the osseous remains from previous excavations made it possible to quantify the material by studying species distribution, using Number of Identifiable Specimens (NISP), and to study bone preservation through weathering analyses. The bones from both old and new excavations were determined to species or family level using the reference collections at the Zoological museum and department of Archaeology and Ancient History at Lund University and the collections at the National Historical Museums in Sweden. The weathering analyses were done according to following criteria: Category 1: Light beige to medium brown, hard and shining surface, no cracks, heavy bones, equivalent to Behrensmeyer [40] stage 0. Category 2: Reddish, dark brown to black-brown, hard and shining surface, no or occasional cracks, heavy bones, Behrensmeyer stages 0–1. Category 3: Light to medium brown, dull, corroded surface, longitudinal cracks, exposure of cancellous bones, light bones. Behrensmeyer stages 1–2. Category 4: Light brown, dull surface, the outer surface is missing due to exfoliation, light bones. Behrensmeyer stages 3–5 (Fig 6).

For weathering analyses on the bones from the new excavation, this was deemed inadequate, due to more extensive bone loss noted on the newly excavated bones (c.f. also the range of category 4 spectrum, Fig 6D and 6E). Consequently, prior to analyses, an additional weathering class was created (called weathering II) as a way of describing the details of the original weathering 4 category. Weathering II is, consequently, a four new sub-categories detailed description of the original weathering I category 4, as follows: Category 5: all sides outer surface erosion less than 50%. Category 6: all sides outer surface erosion more than 50%. Category 7: no remaining surface, all sides average bone loss less than 5 mm. Category 8: no remaining surface, all sides average bone loss more than 5 mm (S1 Fig in S1 File). For illustrative purposes, a Category 9 was also included. This is not an actual weathering category but an illustration to show a complete lack of unburnt bone in a particular trench. Because the bones from the old excavation campaigns were analysed using only 4 weathering categories, comparisons between new and old excavations were done using the first four weathering categories. For intra-excavation comparison, i.e. comparison between the different 2019 trenches, all weathering categories were used.

## Material

This study focuses mainly on the osteological material from three separate excavations of Ageröd I. The material from the 1940s and 1970s excavations of Ageröd I (B, D & HC) amount to a total of 4180 bone fragments determined to species or family level. From the Ageröd I:HC section 299 bone or antler artefacts have also been identified [35], which combined with the 18 bone and antler artefacts from the I:B & D section [21] place the osseous remains from Ageröd I as one of the largest and best-preserved Middle Mesolithic assemblages in northern Europe. When also considering other organic remains, recovered on the early excavations of the site, such as worked wood, resin, pollen and hazelnuts etc., the vast potential of the organic material culture from the site is highlighted and constitutes one of the reasons why this site was selected for re-excavation and analysis.

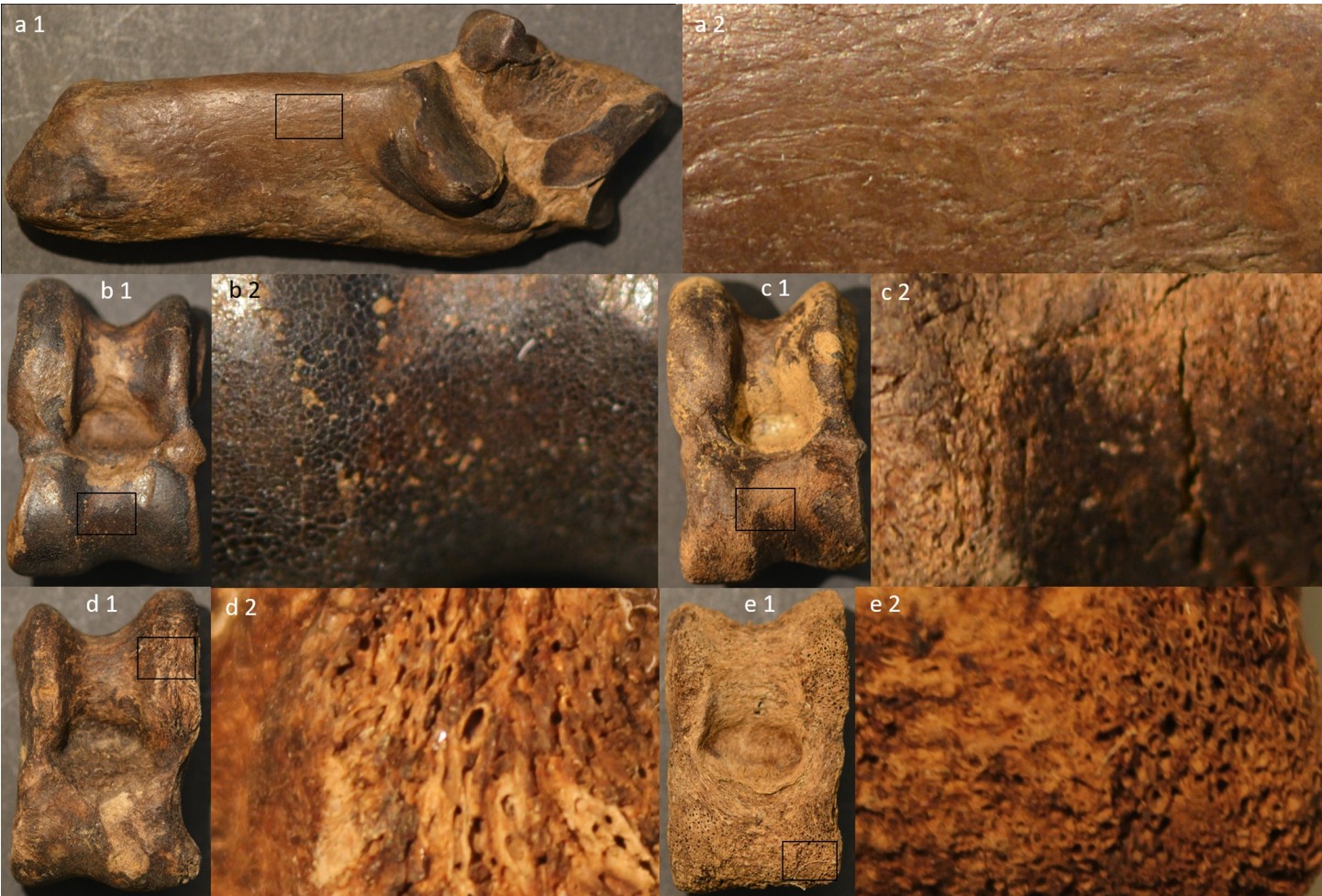

**Fig 6. Weathering I categories exemplified with bones from the 1940s excavation campaign.** a: Wild boar calcaneus weathering category 1, b: Red deer astragalus weathering category 2, c: Red deer astragalus weathering category 3, d: Red deer astragalus weathering category 4 (lower end of category 4 spectrum), e: Red deer astragalus weathering category 4 (upper end of category 4 spectrum). Image created for this publication by the authors (OM and AB).

During the 2019 excavation, an additional 105 bone fragments, including seven bone, tooth or antler artefacts, were also recovered. Of these bone pieces, 61 fragments could be determined to species or family level, which means that the osseous material from Ageröd I encompass a total of 4241 determined bones (Table 1). In addition to the osseous remains, the 2019 excavation revealed additional hazelnut shells and a variety of seeds from different plants, which were recovered by flotation of soil samples from each trench [39]. Furthermore, two pieces of resin were also recovered on the excavation. One of the resin pieces was found inserted in a slotted bone point (S2 Fig in S1 File) and the other one was found with a flint microlith in its middle and bone dust remains on the outside, which suggests that this resin and flint piece had also previously been inserted in a slotted bone point (S3 Fig in S1 File).

For comparison in this paper, only the determined bones from the Ageröd I:HC excavations have been used, which amounts to 3716 bone fragments from the old excavations and 61 fragments from the 2019 excavation. For the weathering analyses, 52 bones from the new excavations were used and 3392 from the old. All bones from the former excavations could not be assigned to an exact location. This is partly because of the old excavation method, where some of the areas of the site were crudely excavated, and partly because metadata, which could have

connected specific bones to specific contexts/areas, have been lost in the years since the excavations. Consequently, not all bones from the old excavations could be included when analysing the preservation condition stratigraphically or spatially. In total, 181 bones from the 1970s zone 2, 519 bones from 1940s zone 1, 553 bones from 1940s zone 2, 872 bones from 1940s zone 3 and 358 bones from zone 4 were included.

## Result

### Abundance of species

Zooarchaeological analyses included recovered bone material from both the 1940s and the 1970s excavations of Ageröd I. The osteological material from the old excavations is extensive and yielded an array of different species, dominated by ungulates. Because the new excavations were limited and only five square meters were excavated, the number of comparable bones are limited. When comparing the species distribution between the old and new excavations the results are, however, comparable in some aspects while they differ more in others (Table 1).

In the HC sections, the biggest difference in species distribution between the old and the new excavation is the lack of smaller fur game animals, e.g. Canidae, Mustelidae and Felici-dade. In the 2019 excavation, bird bones are also completely lacking, which differs from previous excavations from which 32 bird bones representing a minimum of 13 species could be determined. Fish and reptile bones are also missing in the 2019 excavation, but they were rare in the former excavations as well. Regarding the fish, it is, however, difficult to know what the lack of fish bones from the former excavations indicates. No sieving was done on any of the former excavations and, given the difficulties in finding fishbones when not sieving with small mesh sizes [41–44], the lack of fish bones from the former excavations may indicate deficient recovery methods, as opposed to lack of fishing activities. This also fits best with the location of Ageröd I (on the shore of an ancient lake, Fig 1) and a perceived general fish dependency in Southern Scandinavia at the time [45, 46]. To avoid speculation regarding the fishing activities at the site, this could be further investigated using more high resolute methods, e.g. flotating soil samples to look for microscopic parts of fish bones or by studying microwear analyses on stone tools recovered from the site (see e.g. [47]). The species distribution from ungulates is more similar between the different excavations campaigns and only small differences can be observed, indicating that the deterioration had not affected the species abundance of larger mammals (Fig 7).

### Weathering

All appropriate bones (teeth and small bone fragments excluded) from the new and old excavation were subjected to an analysis of weathering following the criteria stated in the method section. Only bones from animal species found on both old and new excavations were compared species by species. Weathering comparisons were also made including all weathering-determinable bone fragments (Fig 8).

The results from the weathering studies give a clear indication that the osseous material has suffered increased diagenesis during the last 70 years. Furthermore, it is also apparent that the accelerated deterioration had started already in the 1970s. Consequently, the osseous remains had gone from a weathering average of around 2.8 in the 1940s (hard and heavy bones with occasional cracks) to 3.4 in the 1970s (lighter bones with larger cracks and exposure of cancellous bone) to in 2019 an average of 3.7 (light and heavily eroded bones, with loss of outer surface). There are some overlaps in bone preservation between the different excavations, i.e. the worst preserved bones from the 1940s are roughly comparable to the best-preserved bones from the 1970s and the worst bones from the 1970s are comparable to the best-preserved

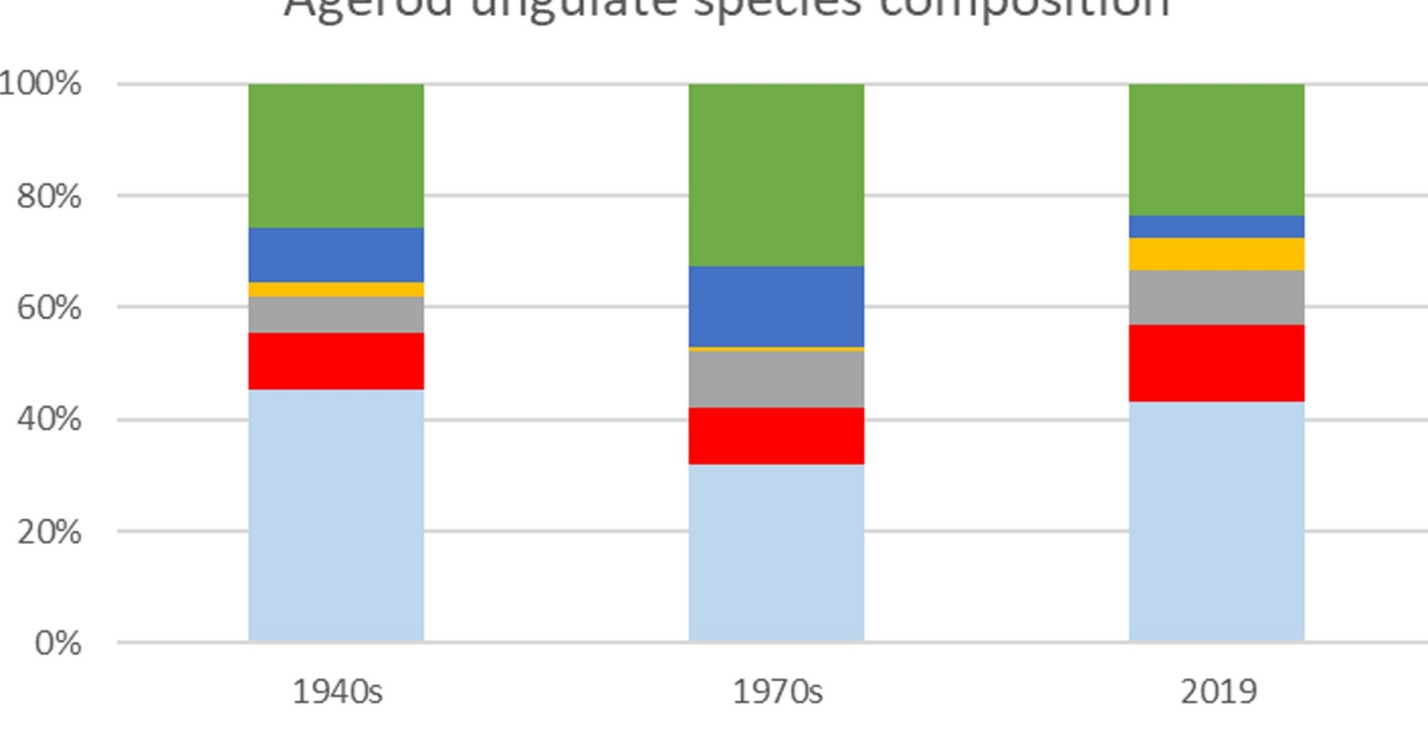

**Fig 7. Species distribution among the ungulates from the different excavation campaigns.** *No specimens belonging to European bison have been determined, whereby it has been assumed that all larger bovid fragments belong to aurochs. N: 1940s = 3019; 1970s = 195; 2019 = 51.

bones from 2019. However, the general trends clearly show a highly accelerated deterioration. When visually comparing the bones from the different decades of excavation (Fig 9), the differences in preservation are also obvious.

Many of the bones from the 2019 excavation are heavily weathered (Fig 9A, 9B and 9D) whereby category 4 was deemed insufficient to use and additional weathering categories were created to study the degradation in the different trenches (see methods). Through this approach, it was possible to establish the average weathering degree on the bones from the different species and to create a 2019 trench-weathering-average (Fig 10), and to study the bone preservation in the different soil layers of each trench (S4 Fig in S1 File).

The bones from trench 201 and 217, both in zone 2, appear to be affected by bone deterioration to a lesser degree than the bones from trench 205, in zone 3. Interestingly, the two trenches with the best preservation were both located in the soil bank in zone 2, where the soil from the drainage ditch had been heaped when the bog was first drained in the early 1900s. Another interesting observation regards trenches 205 and 209, both located on the "lakeside" of the drainage ditch with the lowermost parts of the trenches still in wet condition. Trench 209 is completely lacking unburned bones (although two burnt and fully calcinated bones were found here) while in 205 the weathering degree seems to increase in the deepest and wettest areas of the trench. This trend is not reflected in the bones from the former excavations. When applying the same zone division to the old excavations, where the Ageröd I:HC area is divided into four zones from the driest conditions in zone 1 to the wettest condition in zone 4

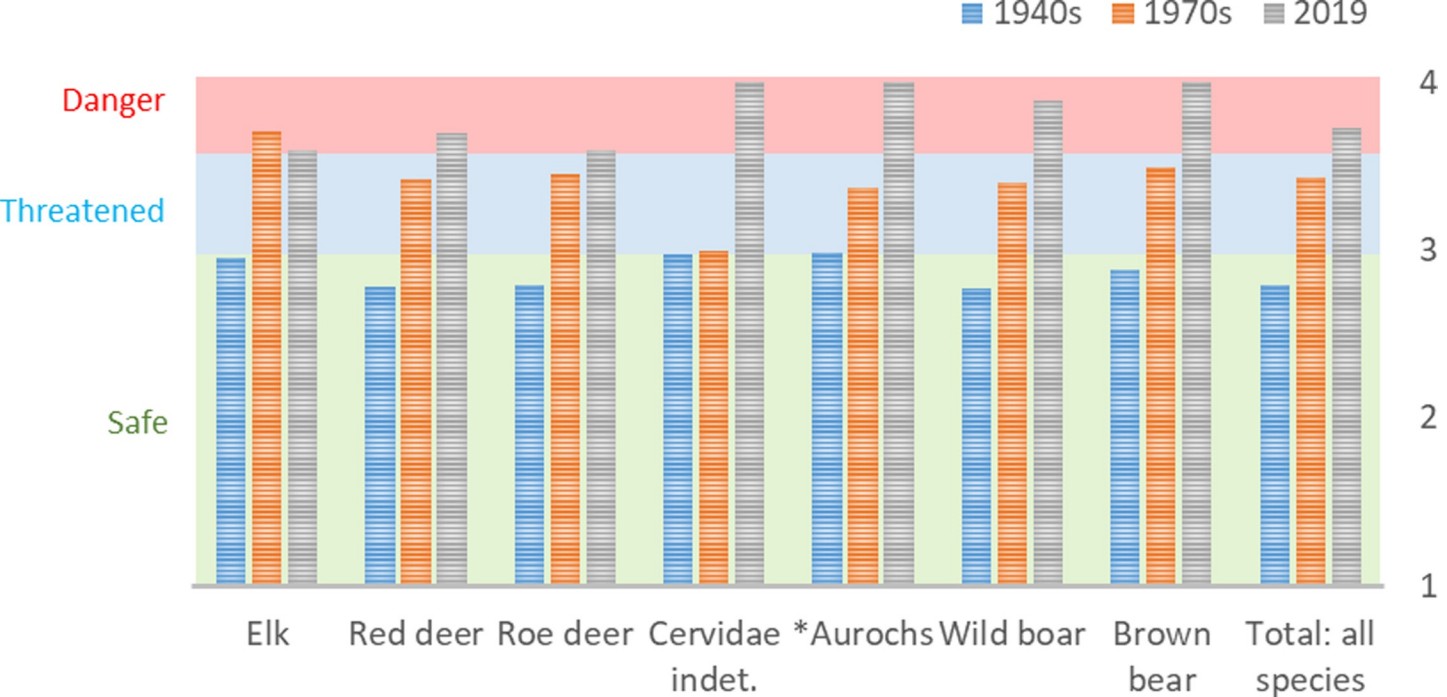

**Fig 8. Mean weathering stages for the bones found during the 1940s, 1970s and the 2019 excavation campaigns, divided into the comparable species and a total for all determinable bone fragments combined.** The severity of degradation increases with higher weathering degree numbers. Weathering degrees <3 are considered low, 3 medium and 4 high. N: 1940s (Elk = 207; Red deer = 1281; Roe deer = 308; Cervidae indet. = 45; *Aurochs = 301; Wild boar = 676; Brown bear = 54; Total all species = 3140); 1970s (Elk = 17; Red deer = 61; Roe deer = 20; Cervidae indet. = 1; *Aurochs = 24; Wild boar = 50; Brown bear = 2; Total all species = 181); 2019 (Elk = 5; Red deer = 20; Roe deer = 5; Cervidae indet. = 3; *Aurochs = 2; Wild boar = 9; Brown bear = 3; Total all species = 52).

(see Fig 4 for zones), the bone preservation was originally best in the wettest and deepest areas of the former trenches (Fig 11 and S5 Fig in S1 File).

The low number of unburnt bones in 2019 zone 3 (with a complete lack of small bones in 205 and no unburnt bones in trench 209) is not caused by any disturbances to the layers as indicated by the fact that the lithic material is present here in comparable numbers to the other 2019 trenches (S6 & S7 Figs in S1 File).

## Discussion

Organic archives are important for our current knowledge of past human activities and cultures and, indeed, of the environment itself [48, 49]. The Mesolithic site Ageröd has been one such archive, were nearly optimal conditions have left a glimpse into the organic material humans used and interacted with, some nine millennia ago. During the last 75 years, this record has suffered from an accelerated deterioration, which threatens its future existence. Yet, nothing "special" has happened to the Ageröd site during this period. In other words, there have been no major constructions in the close vicinity of the site, the area is rather secluded and although the wetland has been drained it has been done so by low-technological methods, i.e. without using mechanical pumps etc. Consequently, the Ageröd site is not highly divergent from most other sites in Northern Europe which have previously shown excellent preservation of archaeological organic material.

The evidence presented here focuses on the osseous remains from the site. By a standard zooarchaeological analysis on both the material from the old excavations and the new, it is possible to detect certain trends. Lately, several Mesolithic sites from Northern Europe have been

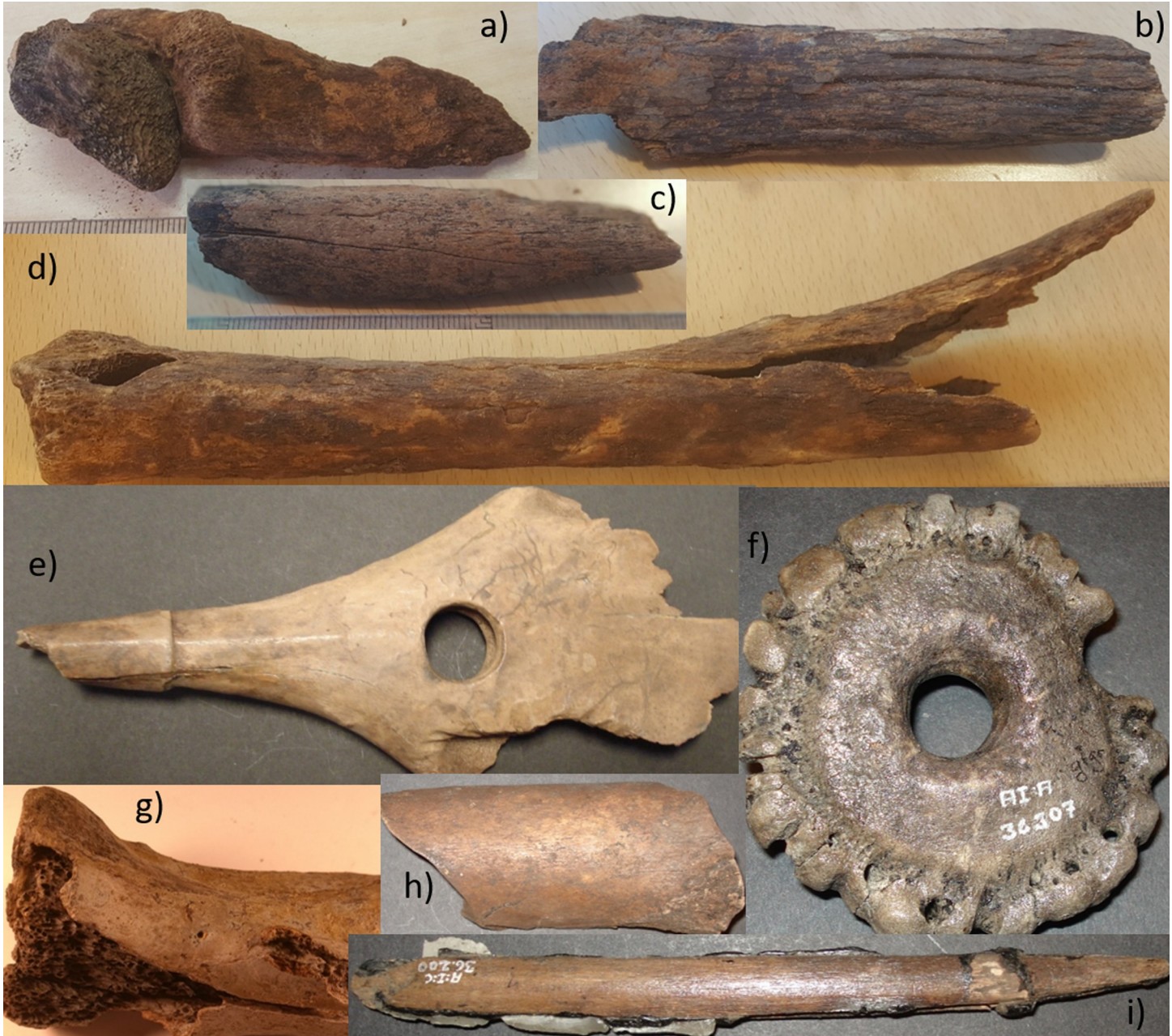

**Fig 9. Organic bone preservation at Ageröd.** a-d bones from 2019; e-i from old excavations. a) astragalus and calcaneus from wild boar found articulated in the transition between white cultural layer and lower peat in trench 205, likely deposited in wet conditions with tendons and ligament still connected, weathering category 8. b) metatarsal from aurochs found in white cultural layer in trench 217, weathering category 6. c) radius diaphysis from elk found in white cultural layer in trench 201, one of the best-preserved bone fragments from the 2019 excavation, weathering category 3. d) tibia from red deer found in white cultural layer of trench 205, weathering category 7. e) drilled and ornated cervid antler from the 1940s, weathering category3. f) "net sinker" made from burr of red deer antler, from the 1940s excavation, weathering category 2. g) scapula from red deer found in the white layer in the 1970s, weathering category 3. h) femur diaphysis from aurochs from the 1940s, weathering category 2. i) slotted bone point from the 1940s, in mint condition with resin and inserted microliths. All photos realised for this publication by the authors (OM and AB).

revisited both physically and by going through old documentation and remains [10, 50–53]. While old excavations are often fraught with different types of problems, which makes certain aspects of the recovered remains difficult to compare, much information can still be obtained. If studying the differences in species distribution between the three excavation campaigns at

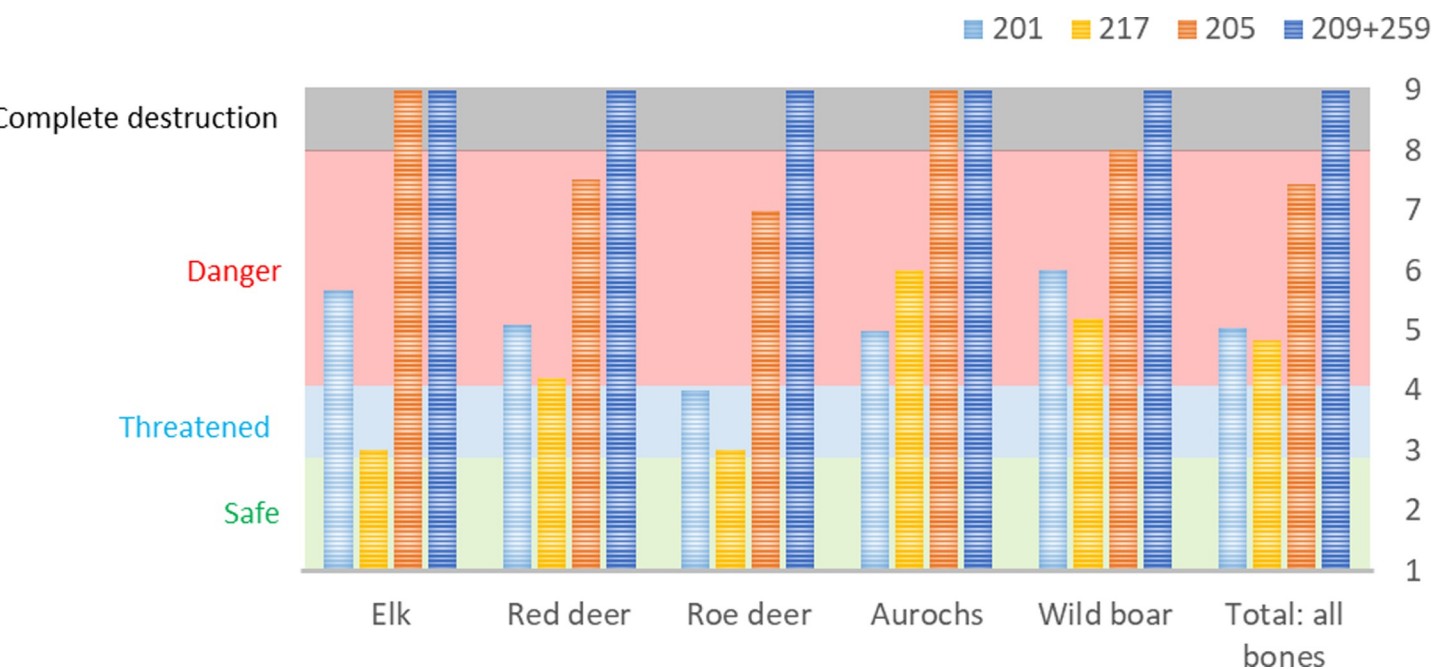

**Fig 10. Mean weathering II values for the ungulates and for all bones combined, divided into the five trenches from the 2019 excavation.** The severity of degradation increases with higher weathering degree numbers. Weathering degrees <3 are considered low, 3 medium and 4–8 high, category 9 indicates lack of preserved unburnt bones. N: Trench 201 (Elk = 3; Red deer = 11; Roe deer = 2; Aurochs = 1; Wild boar = 1; Total all bones = 26); Trench 217 (Elk = 1; Red deer = 5; Roe deer = 1; Aurochs = 1; Wild boar = 5; Total all bones = 17); Trench 205 (Elk = 0; Red deer = 2; Roe deer = 2; Aurochs = 0; Wild boar = 2; Total all bones = 9); Trench 209+259 = 0.

Ageröd it shows, for instance, that few remains from small-bodied species are present in the new excavation, despite water sieving. Because we located our trenches close to the ones from

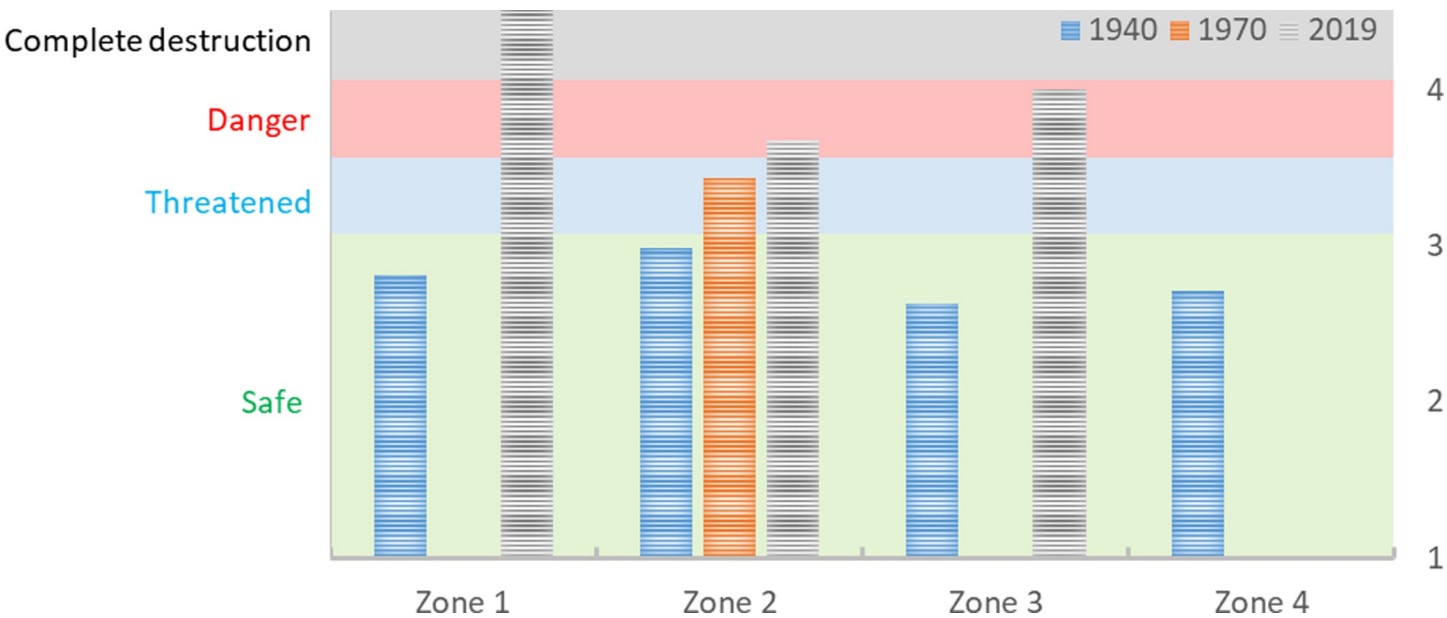

**Fig 11. Comparison between the mean weathering stages on the different excavation campaigns by zone (see Fig 4 for zone specifics).** The severity of degradation increases with higher weathering degree numbers. Weathering degrees <3 are considered low, 3 medium and 4 high. N: 1940s (zone 1 = 541; zone 2 = 590; zone 3 = 964; zone 4 = 358); 1970s (zone 3 = 181); 2019 (zone 2 = 43; zone 3 = 9).

the 1940s and the 1970s (where small-bodied species were found), it is unlikely that our excavation encountered other types of activity areas and a lack of these species is not an effect of depositional variations. If also considering the similarity of the larger ungulates species abundance between the different excavation campaigns in combination with their distinct increase in weathering over time, it suggests that our results are not coincidental. Consequently, it appears as if we are rapidly moving towards a preservation tipping point where the window of opportunity to obtain information from the archaeological organic remains at Ageröd, is rapidly closing.

We set up two test trenches in zone 3 (where the bones from the 1940s excavation had been best preserved) and one of them had no unburnt bones preserved while the material from the other trench showed extensive corrosion leading to massive bone loss on each of the large bones that were still preserved. Here, the bone loss was often more than 5 mm on all sides of the bone, implying that most bones with a cortical bone thickness of less than 1 cm would be completely deteriorated. Also, the corrosion and weathering on the bones increase with increasing trench depth, with the most damaged bones still buried in wet conditions. This requires additional analyses to detect if the added degradation (and a complete loss of unburnt bones in trench 209) depends on lager fluctuations in the water table or if they are caused by the water turning acidic.

One indication of the processes involved can be gained from studying the bone histology, where the presence of oxidised pyrite was detected in bones from the trenches dug in the 2019 excavation campaign, while the histologically analysed bones from the 1940s and 1970s excavations showed only non-oxidised pyrite [39]. Pyrite is an iron sulphide and only forms under anoxic conditions due to the action of sulphate-reducing bacteria. If oxygen is re-introduced into the environment, these pyrite grains will oxidize and turn into iron-hydroxides at the same time as sulphuric acid is produced [54, 55]. The released sulphuric acid will etch and damage the bone, and cause pH levels to drop in the area. Indeed, the fact that oxidised pyrite was only detected in the bones from the 2019 excavations and the bones from previous excavations show only non-oxidised pyrite, suggests that the local soil environment has gone from stable anoxic to oxic conditions [39].

Consequently, it is conceivable that the observed elevated bone destruction in the deepest and wettest parts of trench 205, and the complete bone destruction in trench 209, is caused by the groundwater having turned acidic by sulphuric acid seeping down from the archaeological layers above. Indeed, preliminary results from the excavation suggested a correlation between lower pH levels in the soil with increasing trench depths, when using a pH and moisture soil probe (EAN 2007005725967). These observations were also confirmed by the soil chemical analyses. The two trenches without bone remains (209 and 259) are the most oxygenised and have the lowest pH levels (between pH 4.2 and 5.0), thus displaying the worst soil properties for bone preservation [39], which is a pattern also observed at other sites (e.g. [10]). But, as mentioned above, parts of trench 259 had been disturbed by a former non-registered trench and the results are, consequently, biased.

In trench 205, placed in an area where bones are preserved, but where the bone preservation is worst, the soil analyses show an almost neutral pH-value. However, in trench 205 the lowermost soil sample was extracted between 15–18 cm above the underlying moraine (the moraine is roughly corresponding with the groundwater level) while the soil sample in 209 was extracted only 5 cm above the moraine and thus is directly connected to the groundwater. Furthermore, no bones were recovered from the lowermost 10 cm in trench 205 (where it was wet) and, just as in trench 209 and 259, the soil in 205 had oxidated [39], suggesting that the soil has lost all its buffering against low pH.

These observations are in agreement with the histological analyses. The microstructure of the interior central parts of the cortical bones from the wet conditions in trench 205 shows that they are among the histologically best-preserved bones from 2019 [39], despite suffering from elevated surface etching and corrosion. This demonstrates that the inner cortical microstructure of the bones from the previous best-preserved area, zone 3, is still mostly unaffected by bioerosion and microbial activity [39].

Surface corrosion is also observed on the palaeobotanical record from zone 3. On both water-pepper seeds (*Polygonum hydropiper*) and hazelnut shells (*Corylus avellana)* similar surface corrosion and etching, as detected on the bones in trench 205 (Fig 9A and 9D), is noted (Fig 12). Also, the palaeobotanical analyses show earthworm eggs (*Lumbricus sp.)* in the lower areas of both trench 201 and 209 [39]. Since earthworms cannot live in anoxic layers, this is a further indication of oxygenated soil. More importantly, the inclusion of earthworms will also act to increase soil oxygenation, because of how earthworms loosen up soil with their movements, which will cause further drainage and oxygen admittance, in a (from a perspective of archaeo-organic preservation) downward spiral.

The observed accelerated bone surface destruction in the formerly best-preserved areas of zone 3 and the observed corrosion on palaeobotanical remains from the same area is, given the above-mentioned observations, related to an increased groundwater acidity and fluctuations in the groundwater table. The acidity is likely caused by a combination of acid precipitation and re-oxygenated soils, which cause the pyrite to oxidise and add sulphuric acid to the local groundwater. Without a long-term monitoring program for the site, it is not possible to detect fluctuations in the groundwater level. However, the soil acidity observed in the lowermost areas of trench 209 (and the absence of unburnt bones), combined with the lack of bones in the lowermost 10 cm of trench 205 and the surface corrosion on the bones in the areas stratigraphically above these wet conditions, with increasing degradation with increasing recovery depth (S4 Fig in S1 File), makes it plausible that the bone surface corrosion observed in 205 has been caused by fluctuation in the groundwater table. It is conceivable that water fluctuations have enabled acidic groundwater to corrode the bone surface during high groundwater levels, but for limited periods. If acidic groundwater, with pH levels matching those observed at the bottom of trench 209 (pH 5,2), had been in contact with bones recovered in 205 for extended periods, the bones that were still preserved here would likely have dissolved. This is further strengthened by the lack of small unburnt bones in the trench and the high weathering levels on the remaining large bones, from which more than 5 mm of the outer bone surface has deteriorated.

Another interesting observation is that the best-preserved bones from 2019 were found in the layers under the soil-heap from the drainage ditch. While the ditch itself was full-scale destruction of the archaeology in the area where the ditch was dug, it also appears as if the combination of drainage and added soil seems to have acted to preserve the bone remains. The bones from the trenches situated in the soil bank are affected by an accelerated deterioration, but it is less severe here than in the other areas (Fig 11 and S5 Fig in S1 File). The best-preserved bones from zone 2, under the soil bank, are comparable to the worst preserved bones from the 1970s, which suggest that the combination of a buffering soil on top of the archaeological remains and the removal of what seems to be acidic groundwater, does act to delay bone degradation. However, the offered protection is not optimal and while it delays the deterioration, it does not protect the remains (for more intentional deterioration threat mitigation strategies see e.g. [56–58]. Thereby, merely adding soil on top of threatened cultural remains will only offer a short-term "time-to-act" solution and not long-term protection. Consequently, while the added soil has offered some buffer against degradation, time is running out. Oxygen has entered the cultural layers down into the deepest parts of the trenches, evident

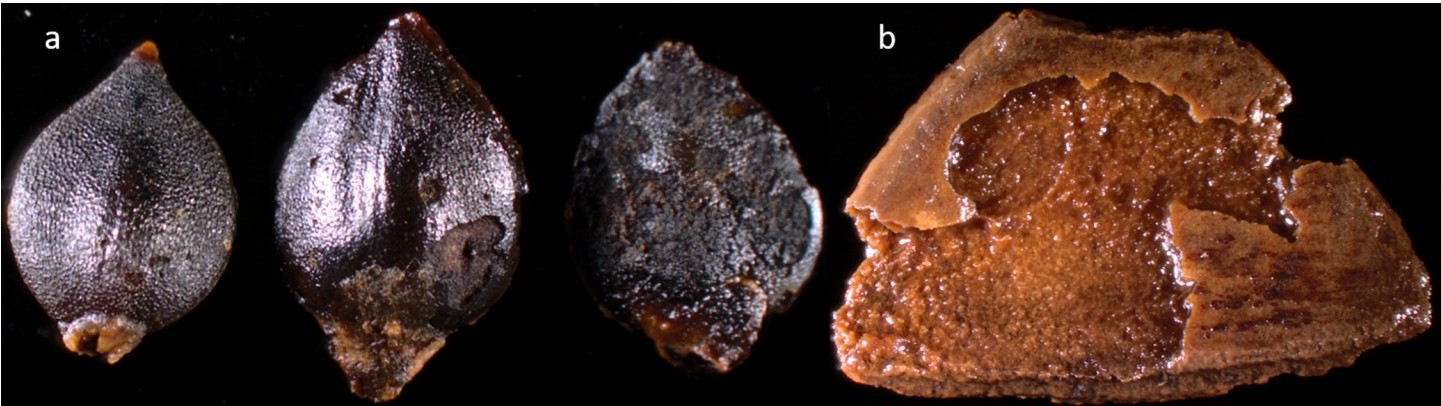

**Fig 12.** Seeds from water-pepper (a) and hazelnut (b) from trench 209 showing signs of corrosion and etching likely caused by an oxygenated and acidic environment. Photography: Santeri Vanhanen, created for this publication.

through both earthworm eggs, large cracks in the peat and soil (caused by the soil dehydration), and soil chemical analyses, which show that the soils are oxidised even in the deepest lying layers [39]. This is causing an accelerated humification and peat degradation process and a non-reversible deterioration of the archaeological organic remains, which, if not stopped or excavated, will cause the organic remains at Ageröd to disappear in a not too distant future.

## Concluding remarks

The organic archaeological remains on Ageröd are threatened by accelerating organic destruction. The best-preserved areas in the 1940s have now become the worst areas for organic preservation. This is related to an acidification of the groundwater in the bog, which might be related to both re-oxygenated soil conditions, causing the release of sulphuric acid, as well as acid precipitation. This raises serious questions about the preservation conditions of similar wetland sites in Northern Europe.

If oxygen is introduced to anoxic layers, through drainage of wetlands, extreme drought leading to dry cracks in the previously wet peat, or indeed, previous archaeological excavations, it might start a reaction where pyrite oxidizes and in the process develops sulphuric acid. If this should happen to layers that will once again become wet, the sulphuric acid might be released into the surrounding water, causing pH-levels in the groundwater to drop. This can be connected to the differences in preservation observed at the different zones of Ageröd. The water in the wet areas might encompass and surround the bones with acidic groundwater, while the acidity released in the dry zones will seep further down in the ground, until reaching the groundwater and thus adding further acidity to the affected basin while preventing the bones in the dry conditions from being "soaked" in acidic water.

The excavation in 2019 has highlighted the problems of organic deterioration. We have shown that the bones are suffering badly from accelerated degradation and that in some areas this has destroyed all bone remains, which only 75 years ago had often been almost pristinely preserved for over 9000 years. The cause for alarm is great and it is likely that the bones even further out in the Ageröd basin have been even more affected, which raises questions of the status of the organic remains not only on Ageröd I but also on e.g. Ageröd V, which had remarkable organic preservation in the excavations in the late 1970s [22]. Further investigations are needed, on both the soil and the organic remains from Ageröd I, but also from other parts of the Ageröd basin.

Furthermore, we have to start investigating the preservation conditions at other sites to find out how modern encroachments, acidification and climate change have affected the organic remains elsewhere. For example, the drainage of wetlands has been widely practised throughout the world to create more arable land and/or improve wood production (cf. e.g. [59, 60]). This has had dire consequences for the preservation of organic remains [61]. Extreme weather situations, with summers of extreme drought (such as the summer of 2018), are likely also responsible for the drying of wetlands and lowering of the groundwater table. In addition, anthropogenically induced climate change does seem to increase the likelihood of extreme weather situations, e.g. both droughts and extreme rainfalls [62]. This, if considered in the light of recent studies suggesting that droughts are expected to drastically increase in the future [63], is cause for alarm concerning the future of ancient organic preservations in wetlands in general.

Our results from Ageröd I are alarming and calls for a large-scale examination of other archaeological sites with organic preservation. The hidden archaeological landscape needs to be re-examined, site-specific deteriorations must be quantified and local prerequisites must be evaluated before appropriate protective actions to secure our ancient heritage can be taken. Furthermore, we need a plan of how to stop or deal with the ongoing accelerated deterioration on a broad international scale and we need to consider if we can and should excavate these threatened sites now, while they still have organic preservation. Furthermore, we need to rethink our approach on managing our archaeological cultural heritage, as our current view in favour of *in situ* preservation [64] is depending on stable soil-environments, which is, evidently, not the case [65]. Given the enormity of this task, we ultimately also need to reconsider the future of our common cultural heritage and the heritage sector needs to adapt its long-term practice [66]. How the protection of threatened sites will be financed along with a change and updating of both national and international heritage legislations are future challenges at hand. The time to act is now and if we wait, we might be in a situation where no matter what actions we take, all organic remains at Ageröd (and possibly at many other sites) will be lost forever.

## Supporting information

**S1 File.**
(PDF)

## Acknowledgments

All necessary permits were obtained for the described study, which complied with all relevant regulations of the County Administrative Board of Scania, Sweden. We would like to thank the three reviewers for their insightful comments. We would also like to thank Lars Larsson for helping us locate the old excavation trenches at Ageröd, for sharing his profound knowledge of the site and for giving us some of his photos from the 1970s excavation. We would also like to thank Arne Sjöström and Santeri Vanhanen for letting us use their figures. We acknowledge and thank the curators at Lund University Historical Museum for permission to analyse and sample bone remains in their care and for letting us use the photos from the 1940s excavation. Lastly, we are grateful for the assistance of our collaborators in the Stone Age for the Future Network and everybody else who volunteered and assisted during our excavation of the site.

## Author Contributions

**Conceptualization:** Adam Boethius.

**Data curation:** Adam Boethius, Mathilda Kjällquist.

**Formal analysis:** Adam Boethius, Ola Magnell.

**Funding acquisition:** Adam Boethius, Jan Apel.

**Investigation:** Adam Boethius, Mathilda Kjällquist, Ola Magnell.

**Methodology:** Adam Boethius.

**Project administration:** Adam Boethius.

**Writing – original draft:** Adam Boethius.

**Writing – review & editing:** Adam Boethius, Mathilda Kjällquist, Ola Magnell, Jan Apel.

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
