## [Decision Letter · Decision Letter 0]

15 May 2020

PONE-D-20-10718

Human encroachment, climate change and the loss of our archaeological organic cultural heritage: Accelerated bone deterioration on Ageröd, a revisited Scandinavian Mesolithic key-site in despair

PLOS ONE

Dear Dr. Boethius,

Thank you for submitting your manuscript to PLOS ONE. After careful consideration, we feel that it has merit but does not fully meet PLOS ONE’s publication criteria as it currently stands. Therefore, we invite you to submit a revised version of the manuscript that addresses the points raised during the review process.

All comments have to be addressed before re-submission.

We would appreciate receiving your revised manuscript by Jun 29 2020 11:59PM. To enhance the reproducibility of your results, we recommend that if applicable you deposit your laboratory protocols in protocols.io, where a protocol can be assigned its own identifier (DOI) such that it can be cited independently in the future. For instructions see: http://journals.plos.org/plosone/s/submission-guidelines#loc-laboratory-protocols

We look forward to receiving your revised manuscript.

Kind regards,

Peter F. Biehl, PhD

Academic Editor

PLOS ONE

Journal Requirements:

2.  In your manuscript, please provide additional information regarding the specimens used in your study. Ensure that you have reported specimen numbers and complete repository information, including museum name and geographic location.

For more information on PLOS ONE's requirements for paleontology and archaeology research, see https://journals.plos.org/plosone/s/submission-guidelines#loc-paleontology-and-archaeology-research.

Additional Editor Comments (if provided):

Your manuscript has now been seen by three referees, whose comments are appended below. You will see from these comments that while the referees find your work of of great interest, they have raised some concerns that must be addressed. In light of these comments, we cannot accept the manuscript for publication, but would be interested in considering a revised version that addresses these concerns.

Reviewers' comments:

Reviewer's Responses to Questions

**Comments to the Author**

1. Is the manuscript technically sound, and do the data support the conclusions?

Reviewer #1: Yes

Reviewer #2: Partly

Reviewer #3: Yes

2. Has the statistical analysis been performed appropriately and rigorously? 

Reviewer #1: Yes

Reviewer #2: N/A

Reviewer #3: Yes

3. Have the authors made all data underlying the findings in their manuscript fully available?

Reviewer #1: Yes

Reviewer #2: Yes

Reviewer #3: Yes

4. Is the manuscript presented in an intelligible fashion and written in standard English?

Reviewer #1: Yes

Reviewer #2: No

Reviewer #3: Yes

5. Review Comments to the Author

Reviewer #1: This is a very important if deeply disturbing article, and parallels observations in the circumpolar north and high elevations. The news is very bad, but if it sparks large scale responses (and lots of rescue excavation) we have a chance to save something for the future. Very nice work, very frightening results. This needs immediate publication.

Reviewer #2: While I fully agree with the sentiment and broad conclusions of this paper, its presentation is currently not suitable for publication. It needs heavy and consistent language editing and some parts should, I suggest, be revised. This is true of both the conclusions drawn in relation to the data at hand - some statements are rather strong given the inherent uncertainties - while others are missed out on: this is particularly urgent with reagrd to the need for revised heritage management and financing aspects for 'climate rescue excavations'.

I've tried to help with some of these aspects, so please see the comments in the annotated pdf.

Finally, I just noticed that also the title might need revising (it's 'at' not 'on' Ageröd)

Reviewer #3: Review PLOS ONE paper Boethius et al. Human encroachment…

Good article with important topic, but sometimes too focused on Swedish and/or Mesolithic material and publications. Quite a bit of work on in situ site preservation and changing preservation conditions has been carried out, particularly in the UK, DK, NL and Norway - some Mesolithic, many other periods, all relevant in this context. Some ref suggestions below.

minor language and typographical errors corrected (-> = change to)

JoAS: Journal of Archaeological Science; CMAS: Conservation and Management of Archaeological Sites

l. 22: unique,

l. 30: today if compared – delete if

l. 32: has not been -> has rarely been (there are refs on next page to published analyses)

l. 39: ago,

l. 96: Photography’s -> Photographs

l. 97: profile -> section

l. 119 70s -> 1970s

l. 130: was done by -> was carried out by

l. 138: fixpoints in -> fixpoints from

l. 147: soil bank from when the ditch was dug and maintained throughout the last decades. -> soil bank of excavated ditch material from its establishment and maintenance.

l. 147: arbitrary -> arbitrarily

l. 153: excavation,

l. 154: layers -> deposits

l. 155: is -> are

l. 156: by layer, and all layers -> stratigraphically, and all deposits

l. 160, 165, 167, 290, 292, 372, 431: cultural layer(s) -> archaeological deposit(s)

l. 161: layers -> deposits

l. 164: Profiles -> Sections

l. 165: show -> shows

l. 178: surface,

l. 181, 182, 183: stage -> stages

l. 193: categories -> sub-categories

l. 212: constitute -> constitutes

l. 214: excavation,

l. 217: remains,

l. 238, 240: are -> is

Table 1, l. 290 (and all other use): Elk -> Moose (Elk= Cervus canadensis; Moose = Alces alces)

l. 248: excavation,

l. 252, 306: indicate -> indicates

l. 257, 258: is -> are

l. 265: excavation -> excavations

l. 272, 305, 325: increase -> increases

l. 298: was created -> were created

l. 307: appears -> appear

l. 309: delete when

l. 311: trench -> trenches

l. 319: numbers -> number

l. 331: were -> where

l. 338: who -> which

l. 340: focus -> focuses

l. 341: delete comma after analysis

l. 344: add reference, Boreham et al. 2011 JoAS 38, 2833-57

l. 362: form -> forms

l. 377: add references to published material, eg. Gregory & Matthiesen 2012 CMAS 14, 479-86, Boreham et al. 2011 - possibly more; you are not the only ones to have come across this problem

l. 426: cultural -> archaeological

l. 428: remains. ->add references: Martens et al. 2016 CMAS 18,8-29; Martens 2017 ARC 32.2, 123-140; Tjelldén et al. 2016 CMAS 18, 126-138

Fig. 4: Zone boarders -> Zone borders

Fig. 7, Fig. 10: Elk -> Moose

6. PLOS authors have the option to publish the peer review history of their article (what does this mean?). If published, this will include your full peer review and any attached files.

Reviewer #1: Yes: Thomas H McGovern

Reviewer #2: Yes: Felix Riede

Reviewer #3: Yes: Vibeke Vandrup Martens

---

## [Author Response · Author response to Decision Letter 0]

10 Jun 2020

Reviewer #1: This is a very important if deeply disturbing article, and parallels observations in the circumpolar north and high elevations. The news is very bad, but if it sparks large scale responses (and lots of rescue excavation) we have a chance to save something for the future. Very nice work, very frightening results. This needs immediate publication.

Answer:Thanks

Reviewer #2: While I fully agree with the sentiment and broad conclusions of this paper, its presentation is currently not suitable for publication. It needs heavy and consistent language editing and some parts should, I suggest, be revised. This is true of both the conclusions drawn in relation to the data at hand - some statements are rather strong given the inherent uncertainties - while others are missed out on: this is particularly urgent with reagrd to the need for revised heritage management and financing aspects for 'climate rescue excavations'.

Answer: Toned down the parts deemed as too strong and included a small section of heritage management issues in the conclusion. Changed language according to suggestions by the reviewers and have done a final language revision after that. 

I've tried to help with some of these aspects, so please see the comments in the annotated pdf.

Answer: Manuscript revised following comments in first submitted manuscript pdf, the changes are shown in the marked up copy of the manuscript.

Regarding the comment about the referred manuscript paper (line 362 & 384), we agree that it is not optimal to cite an unfinished manuscript. We have, however, stated the results from this paper in a clear way as to show that there can be no misinterpretations. If we were to also include and show them here it would take the focus from what we are presenting here and because the manuscript paper includes histology, soil chemistry, collagen preservation and palaeobotanical analyses, it would double the length of this paper (which would not be optimal or reader friendly). 

Finally, I just noticed that also the title might need revising (it's 'at' not 'on' Ageröd)

Answer: Fixed

Reviewer #3: Review PLOS ONE paper Boethius et al. Human encroachment…

Good article with important topic, but sometimes too focused on Swedish and/or Mesolithic material and publications. Quite a bit of work on in situ site preservation and changing preservation conditions has been carried out, particularly in the UK, DK, NL and Norway - some Mesolithic, many other periods, all relevant in this context. Some ref suggestions below.

minor language and typographical errors corrected (-> = change to)

JoAS: Journal of Archaeological Science; CMAS: Conservation and Management of Archaeological Sites

l. 22: unique, Answer: Done

l. 30: today if compared – delete if Answer: Done

l. 32: has not been -> has rarely been (there are refs on next page to published analyses) Answer: Done

l. 39: ago, Answer: Done

l. 96: Photography’s -> Photographs Answer: Done

l. 97: profile -> section Answer: Done

l. 119 70s -> 1970s Answer: Done and also updated elsewhere

l. 130: was done by -> was carried out by Answer: Done

l. 138: fixpoints in -> fixpoints from Answer: Done

l. 147: soil bank from when the ditch was dug and maintained throughout the last decades. -> soil bank of excavated ditch material from its establishment and maintenance. Answer: Done

l. 147: arbitrary -> arbitrarily Answer: Done

l. 153: excavation, Answer: Done

l. 154: layers -> deposits Answer: Done

l. 155: is -> are Answer: Done

l. 156: by layer, and all layers -> stratigraphically, Answer: and all deposits changed to “stratigraphically, and all depositional layers

l. 160, 165, 167, 290, 292, 372, 431: cultural layer(s) -> archaeological deposit(s). Answer: It is misleading to use the term deposits when we are discussing stratigraphically deposited layers. By more clearly stating “depositional layers” (see from previous comment) we think the term cultural layer or archaeological layer is better

l. 161: layers -> deposits Answer: Done

l. 164: Profiles -> Sections Answer: Done

l. 165: show -> shows Answer: Done

l. 178: surface, Answer: Done

l. 181, 182, 183: stage -> stages Answer: Done

l. 193: categories -> sub-categories Answer: Done

l. 212: constitute -> constitutes Answer: Done

l. 214: excavation, Answer: Done

l. 217: remains, Answer: Done

l. 238, 240: are -> is Answer: Done

Table 1, l. 290 (and all other use): Elk -> Moose (Elk= Cervus canadensis; Moose = Alces alces). Answer: No, we use the Euroasian term and not the North American

l. 248: excavation, Answer: Done

l. 252, 306: indicate -> indicates Answer: Done

l. 257, 258: is -> are Answer: Done

l. 265: excavation -> excavations Answer: Done

l. 272, 305, 325: increase -> increases Answer: Done

l. 298: was created -> were created Answer: Done

l. 307: appears -> appear Answer: Done

l. 309: delete when Answer: Done

l. 311: trench -> trenches Answer: Done

l. 319: numbers -> number Answer: Done

l. 331: were -> where Answer: Done

l. 338: who -> which Answer: Done

l. 340: focus -> focuses Answer: Done

l. 341: delete comma after analysis Answer: Done

l. 344: add reference, Boreham et al. 2011 JoAS 38, 2833-57 Answer: Done

l. 362: form -> forms Answer: Done

l. 377: add references to published material, eg. Gregory & Matthiesen 2012 CMAS 14, 479-86, Boreham et al. 2011 - possibly more; you are not the only ones to have come across this problem. Answer: Added “, which is a pattern also observed elsewhere (e.g. Borehamn et al., 2011)”

l. 426: cultural -> archaeological Answer: Done

l. 428: remains. ->add references: Martens et al. 2016 CMAS 18,8-29; Martens 2017 ARC 32.2, 123-140; Tjelldén et al. 2016 CMAS 18, 126-138 Answer: Added”( (for more intentional deterioration threat mitigation strategies see e.g. Martens et al. 2016; Martens 2017; Tjelldén et al. 2016)”

Fig. 4: Zone boarders -> Zone borders Done

Fig. 7, Fig. 10: Elk -> Moose No, we use Euroasian term

---

## [Decision Letter · Decision Letter 1]

30 Jun 2020

Human encroachment, climate change and the loss of our archaeological organic cultural heritage: Accelerated bone deterioration at Ageröd, a revisited Scandinavian Mesolithic key-site in despair

PONE-D-20-10718R1

Dear Dr. Boethius,

We’re pleased to inform you that your manuscript has been judged scientifically suitable for publication and will be formally accepted for publication once it meets all outstanding technical requirements.

Kind regards,

Peter F. Biehl, PhD

Academic Editor

PLOS ONE

Additional Editor Comments (optional):

Please make minor corrections listed by reviewer 3 before submission.

Reviewers' comments:

Reviewer's Responses to Questions

**Comments to the Author**

1. If the authors have adequately addressed your comments raised in a previous round of review and you feel that this manuscript is now acceptable for publication, you may indicate that here to bypass the “Comments to the Author” section, enter your conflict of interest statement in the “Confidential to Editor” section, and submit your "Accept" recommendation.

Reviewer #1: All comments have been addressed

Reviewer #2: All comments have been addressed

Reviewer #3: All comments have been addressed

2. Is the manuscript technically sound, and do the data support the conclusions?

Reviewer #1: Yes

Reviewer #2: Yes

Reviewer #3: Yes

3. Has the statistical analysis been performed appropriately and rigorously? 

Reviewer #1: Yes

Reviewer #2: N/A

Reviewer #3: Yes

4. Have the authors made all data underlying the findings in their manuscript fully available?

Reviewer #1: Yes

Reviewer #2: Yes

Reviewer #3: Yes

5. Is the manuscript presented in an intelligible fashion and written in standard English?

Reviewer #1: Yes

Reviewer #2: Yes

Reviewer #3: Yes

6. Review Comments to the Author

Reviewer #1: This is an excellent and timely paper, revisions make it stronger. This problem is widespread, but most focus has been on circumpolar (e.g. Svalbard, Greenland, Alaska) or high elevation cases. This case is particularly troubling and calls into question the dominant rubric of preservation in situ without excavation.

Reviewer #2: Thanks for addressing all outstanding concerns in your revision. I look forward to seeing this paper published.

Reviewer #3: The authors have included almost all suggestions from previous review. However, a few more language errors were discoverd during this reading.

Two general comments; PLOS ONE is a North American publication channel. I therefore recommend that Moose is used for Alces Alces, not Elk (even if that is the Eurasian term).

Archaeological deposits translates into Swedish kulturlager. Depositional layers is a strange term to use (e.g. l. 160).

l. 164: trench -> trenches

l. 168, 170: archaeological white cultural layer -> white archaeological deposit (or layer, if you must)

l. 184: analysis -> analyses

l. 185: where -> were

l. 187: available upon request to the museum -> avalable at the museum upon request/ avalable upon request posed to the museum

l. 198: criteria's -> criteria

l. 200: equivalent with -> equivalent to

l. 233: reasons for why -> reasons why

l. 264 (and consitently): Elk -> Moose (or add note about Elk/Moose)

l. 369: organic preservation -> preservation (organic comes later in same sentence)

l. 414: worst soil properties for organic preservation -> worst soil properties for bone preservation (many organics like acidic environments - wood, leather, textile - but bone certainly doesn't)

l. 415: Borehamn -> Boreham

l. 432 move Latin name after seeds and shells (to increase legibility)

l. 437 etching, -> etching

l. 455: periods,

l. 483: condition -> conditions; on -> of (rewrite sentence: This raises serious questions about the preservation conditions of similar wetland sites in Northern Europe)

l. 487: were pyrite oxidize -> where pyrite oxidizes; develop -> develops

l. 488: ones -> once; wet,

l. 492: and are thus -> and thus

l. 506: These have -> This has

l. 514: calls -> call

l. 522 add reference: McGovern, 2018 (Thomas H. McGovern 2018. Burning Libraries: A Community Response. CMAS, 20:4, 165-174, DOI: 10.1080/13505033.2018.1521205)

l. 537: 4letting -> letting

7. PLOS authors have the option to publish the peer review history of their article (what does this mean?). If published, this will include your full peer review and any attached files.

Reviewer #1: **Yes: **Thomas H McGovern

Reviewer #2: **Yes: **Felix Riede

Reviewer #3: **Yes: **Vibeke Vandrup Martens

---

## [Editor Report · Acceptance letter]

6 Jul 2020

PONE-D-20-10718R1 

Human encroachment, climate change and the loss of our archaeological organic cultural heritage: Accelerated bone deterioration at Ageröd, a revisited Scandinavian Mesolithic key-site in despair 

Dear Dr. Boethius:

I'm pleased to inform you that your manuscript has been deemed suitable for publication in PLOS ONE. Congratulations! Your manuscript is now with our production department. 

Kind regards, 

on behalf of

Dr. Peter F. Biehl 

Academic Editor

PLOS ONE